# Gasdermin D-mediated neutrophil pyroptosis drives inflammation in psoriasis

**Jian Liu[1†], YuYing Jiang[2†], ZiYue Diao[1†], DanDan Chen[1], RuiYuan Xia[1], BingWei Wang[3*], Shuo Yang[2*], ZhiQiang Yin[1*]**

[1]Department of Dermatology, First Affiliated Hospital of Nanjing Medical University, Nanjing, China; [2]Department of Immunology, State Key Laboratory of Reproductive Medicine and Offspring Health, Jiangsu Key Lab of Cancer Biomarkers, Prevention and Treatment, Collaborative Innovation Center for Personalized Cancer Medicine, Gusu School, The Affiliated Wuxi People's Hospital of Nanjing Medical University, Wuxi People's Hospital, Wuxi Medical Center, Nanjing Medical University, Nanjing, China; [3]Department of Pharmacology, Nanjing University of Chinese Medicine, Nanjing, China

**\*For correspondence:**
bingweiwang@njucm.edu.cn
(BWW);
shuoyang01@njmu.edu.cn (SY);
yinzhiqiang@njmu.edu.cn (ZQY)

[†]These authors contributed equally to this work

**Competing interest:** The authors declare that no competing interests exist.

## eLife Assessment

This is a **valuable** study regarding the role of gasdesmin D in experimental psoriasis. The study contains **solid** evidence for such a role, involving neutrophils, from murine models of skin inflammation, as well as correlative data of elevated gasdermin D expression in human psoriatic skin. The findings will be of interest to researchers trying to unravel pathways of skin inflammation.

**Abstract** Psoriasis is a multifactorial immune-mediated inflammatory disease. Its pathogenesis involves abnormal accumulation of neutrophils and T-cell-related abnormalities. Pyroptosis is a type of regulated cell death associated with innate immunity, but its role in psoriasis is unclear. In this study, we found that *gasdermin D (GSDMD)* is higher in human psoriatic skin than that in normal skin, and in imiquimod-induced psoriasis-like mouse skin, the expression of *Gsdmd* was most significantly altered in neutrophils and *Il1b* was also mainly expressed in neutrophils. Immunohistochemical staining of serial sections of skin lesions from psoriasis patients and healthy control also showed that GSDMD expression is higher in psoriasis lesion, especially in neutrophils. *Gsdmd* deficiency mitigates psoriasis-like inflammation in mice. GSDMD in neutrophils contributes to psoriasis-like inflammation, while *Gsdmd* depletion in neutrophils attenuates the development of skin inflammation in psoriasis and reduces the release of the inflammatory cytokines. We found that neutrophil pyroptosis is involved in and contributes to psoriasis inflammation, which provides new insights into the treatment of psoriasis by targeting neutrophil pyroptosis.

## Introduction

Psoriasis is an autoimmune disease that involves multiple systems and manifests primarily as plaques and scaling. The incidence of psoriasis displays significant variation across populations, with estimates ranging from 0.47% in China to 8.5% in Norway, and 3.0% among adults in the United States (*Pezzolo and Naldi, 2020*; *Armstrong et al., 2021*; *Ding et al., 2012*). The etiology of psoriasis is multifactorial and involves a complex interplay of genetic, environmental, infectious, and lifestyle factors.

Previous studies have suggested that the pathogenesis of psoriasis is related to the involvement of various cells, including immune cells, keratinocytes, and stromal cells, and the underlying mechanism is complex (*Guo et al., 2023*). The participation of immune cells remains a research focus. Most biologics utilized to treat psoriasis presently target the immune pathway. Current studies suggest that the interleukin 23 (IL-23)/T-helper 17 (Th17) immune axis is the primary immune pathway involved in psoriasis pathogenesis (*Girolomoni et al., 2017*). Th17 cells play a significant role in the pathogenesis of many autoimmune and inflammatory diseases. Th17-derived proinflammatory cytokines, such as IL-17A, IL-17F, and IL-22, are critical in the development of psoriasis (*Li et al., 2020*). Studies have shown that IL-17A contributes to both an adaptive immune circuit involving specific T-cell subsets and an innate axis between keratinocytes and neutrophils (*Reich et al., 2015*). Except for Th17 cells, neutrophils within the skin are a hallmark feature of psoriasis. Neutrophils accumulate in the dermis and epidermis, resulting in the formation of Kogoj pustules and Munro microabscesses (*Schön et al., 2017*). Neutrophils are increased in the skin and blood of patients with psoriasis (*Rodriguez-Rosales et al., 2021*), and promote psoriasis by secreting proinflammatory mediators and proteases through degranulation or forming neutrophil extracellular traps (*Wang and Shi, 2023*; *Chen et al., 2021*; *Shao et al., 2019*). Nevertheless, the mechanisms by which neutrophils promote and exacerbate inflammation in psoriasis remain under investigation.

Pyroptosis is a type of regulated cell death triggered by the inflammasome and mediated by the gasdermin family (*Galluzzi et al., 2018*). Gasdermin D (GSDMD) is currently the most extensively researched member of the gasdermin family. After perturbations of extracellular or intracellular homeostasis related to innate immunity, the classical and non-classical pathways activate caspase 1 and caspase 4/5/11 (caspase 4/5 in humans and caspase 11 in mice) (*Kesavardhana et al., 2020*), and the activated caspases cleave GSDMD into its N-terminal and C-terminal fragments (*Kovacs and Miao, 2017*). The N-terminus of GSDMD forms a transmembrane pore, which releases inflammatory mediators (*Evavold et al., 2018*; *Heilig et al., 2018*) and results in proinflammatory cell death (*Banerjee et al., 2018*). In recent years, there has been a focus on the role of pyroptosis in infections, tumors, and autoimmune diseases (*You et al., 2022*; *Fang et al., 2020*; *Yu et al., 2021*; *Jiang et al., 2022*) due to its potential to cause inflammatory diseases when occurring excessively.

While previous studies have showed that the expression of GSDMD is increased in lesions of patients with psoriasis (*Nowowiejska et al., 2023*), and the pyroptosis of keratinocytes plays a role in inflammation seen in psoriasis (*Lian et al., 2023*). However, the exact contribution of immune cell pyroptosis to the development of psoriasis remains unclear. Therefore, our study aimed to investigate the role of GSDMD-mediated immune cell pyroptosis in psoriasis.

## Results

### The GSDMD-mediated pyroptosis is activated in psoriasis

By searching the GEO database, we found that *GSDMD* is higher in human psoriatic skin than that in normal skin, and the similar trend was seen in imiquimod (IMQ)-induced psoriasis model mice compared to controls (*Figure 1a*). Immunohistochemical staining of serial sections of skin lesions from psoriasis patients and healthy control also showed that GSDMD expression is higher in psoriasis patients (*Figure 1b*). To further elucidate the presence of pyroptosis in psoriasis, we induced a psoriasis-like mouse model by applying IMQ (*Figure 1c*). The lesions of the mice were collected for western blotting, our results indicated that the IMQ-treated group not only has elevated expression of caspase 1 and GSDMD relative to controls, but also shows functional cleavage of 30KD GSDMD N fragment (*Figure 1d and e*). Hence, the activation of pyroptosis-associated molecules was found in psoriasis lesions.

### GSDMD deficiency mitigates psoriasis-like inflammation in mice

Given the particular elevation of GSDMD levels in psoriasis, we questioned whether GSDMD contributed to psoriasis pathogenesis. We treated WT and GSDMD-deficient mice with IMQ to induce psoriasis (*Figure 1c*). We found that the psoriasis area and severity index (PASI) scores were significantly lower in *Gsdmd*[-/-] mice than WT after 4 days of IMQ application (*Figure 2a and b*). Moreover, indicators to assess the severity of psoriasis, including acanthosis, parakeratosis, and microabscess formation as well as infiltration of leukocytes, were less pronounced in *Gsdmd*[-/-] mice compared with WT

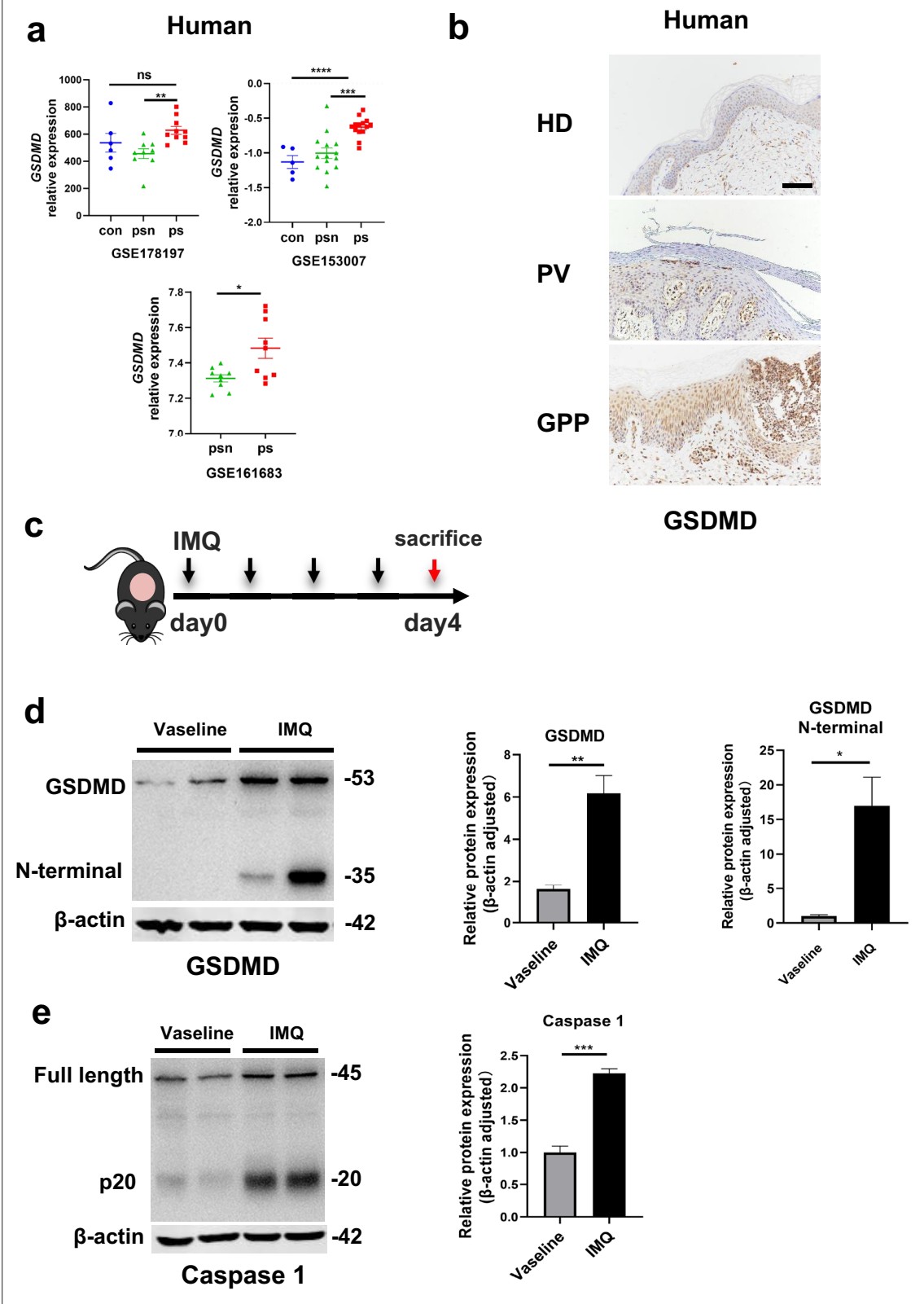

**Figure 1.** The GSDMD-mediated pyroptosis is activated in psoriasis. (**a**) Expression of *GSDMD* in patients with psoriasis patients and healthy people from GSE178197, GSE161683, and GSE153007. (**b**) Representative immunohistochemical staining of GSDMD in sections of skin tissue from healthy individuals and patients with psoriasis (n=3–5, each group). Scale bar = 100 μm. (**c**) Schematic representation of the IMQ-induced psoriasis mouse model. (**d**) Representative images and statistical analysis of western blot analysis showing the expression of GSDMD and its N-terminal fragments in

*Figure 1 continued on next page*

*Figure 1 continued*
dorsal skin of WT mice treated with vehicle or IMQ at day 4 (n=4). (**e**) Representative images and statistical analysis of western blot analysis showing the expression level of pro-caspase 1 and caspase 1 in dorsal skin of WT mice treated with vehicle or IMQ at day 4. psn, psoriasis non-lesional skin; ps, psoriasis skin; con, control; HD, healthy donor; GPP, generalized pustular psoriasis; IMQ, imiquimod; WT, wild-type. Error bars show mean ± SEM. *p<0.05, **p<0.01, ***p<0.001, ***p≤0.0001. Data are representative of three independent experiments for (**b, d, e**).

The online version of this article includes the following source data for figure 1:

**Source data 1.** PDF file containing original western blots for *Figure 1d and e*, indicating the relevant bands and treatments.

**Source data 2.** Original files for western blot analysis displayed in *Figure 1d and e*.

(*Figure 2c*). We verified the absence of GSDMD in the skin of *Gsdmd*[-/-] mice by western blotting and found that after induction of psoriasis in *Gsdmd*[-/-] mice, the expression and activation of caspase 1, an upstream molecule of pyroptosis, were decreased (*Figure 2d*). Levels of the proinflammatory cytokines in protein and mRNA were also examined in the skin using enzyme-linked immunosorbent assay (ELISA) and quantitative PCR, respectively. Our results showed that the mRNA expression of *Il17a*, *Tnfa*, *Il6*, *Il1b*, and *Il18* were substantially higher in the WT mice than in the *Gsdmd*[-/-] mice, and protein expression of IL-1β and IL-6 also significantly decreased in the *Gsdmd*[-/-] mice (*Figure 2e and f*). These results demonstrated that the absence of GSDMD ameliorated IMQ-induced psoriasiform dermatitis in mice.

## Neutrophils undergo GSDMD-mediated pyroptosis in psoriasis

Previous studies have shown that pyroptosis occurs in psoriasis (*Lian et al., 2023*; *Deng et al., 2019*). Moreover, neutrophils are critical to the pathogenesis of psoriasis, which is usually characterized by a high accumulation of neutrophils both in lesions as well as in blood (*Chiang et al., 2019*). Therefore, we wanted to clarify whether neutrophils undergo pyroptosis in psoriasis. First, we searched the GEO database and found that psoriasis patients' neutrophils showed higher expression of GSDMD (*Figure 3a*). Serial sections of skin lesions from psoriasis patients were immunohistochemically stained with CD66b and GSDMD antibodies. We further found that GSDMD was prominently expressed in neutrophils, especially in Munro's microabscesses, which were indicated by CD66b in the patients' skin of both psoriasis vulgaris and pustular psoriasis (*Figure 3b*). Through database search and analysis, we found that in psoriasis-like mouse skin, the expression of *Gsdmd* was most significantly altered in neutrophils (*Figure 3c*). Then, we carried out immunofluorescent analysis of GSDMD in the psoriatic skin tissues of WT mice and GSDMD-deficient mice used as a negative control. Lymphocyte antigen 6 (Ly6G), a classical neutrophil marker (*Ding et al., 2022*; *Xie et al., 2020*), was found to be mainly expressed in neutrophils in the skin of IMQ-induced psoriasis-like mice (*Figure 3—figure supplement 1a*). Therefore, we chose Ly6G as a marker for mouse neutrophils in our experiments. In line with GEO data, immunofluorescence staining showed that GSDMD was more strongly expressed in the mice skin of IMQ applied group than control, including the significant localization of GSDMD in neutrophils (*Figure 3d*). GSDMD is an effector molecule that mediates pyroptosis. We then found that, compared to the control group, neutrophils infiltrating in IMQ-induced psoriasis-like tissue display a higher expression of pyroptosis-related genes through analyzing the publicly available single-cell transcriptomic data (GSE165021) (*Figure 3e*). These results highlight the role of neutrophil pyroptosis in the progression of psoriasis. To further confirm that these neutrophils underwent pyroptosis, we used flow cytometry (FCM) to stain PI to confirm neutrophil pyroptosis and performed TUNEL staining on mouse skin after IMQ application to exclude other cell deaths such as apoptosis (*Figure 3f*, *Figure 3—figure supplement 1b*); these data suggest that GSDMD-mediated pyroptosis exists in neutrophils infiltrating into psoriatic lesions as shown more PI staining but not change in TUNEL staining in neutrophils.

## GSDMD depletion in neutrophils attenuates the development of skin inflammation in psoriasis

To investigate the specific effect of neutrophils in psoriasis, we bred *Gsdmd*[fl/fl]; *S100a8*[cre/+](cKO) mice to make conditionally knockout GSDMD in neutrophils. There are many previous studies using S100a8-Cre mice to knock out neutrophil-specific genes (*Stackowicz et al., 2021*; *Abram et al., 2013*). And we found that *S100a8* is mainly expressed in neutrophils and slightly in macrophages in psoriasis-like mouse skin (*Figure 3—figure supplement 1c*). Therefore, we believe that the knockout

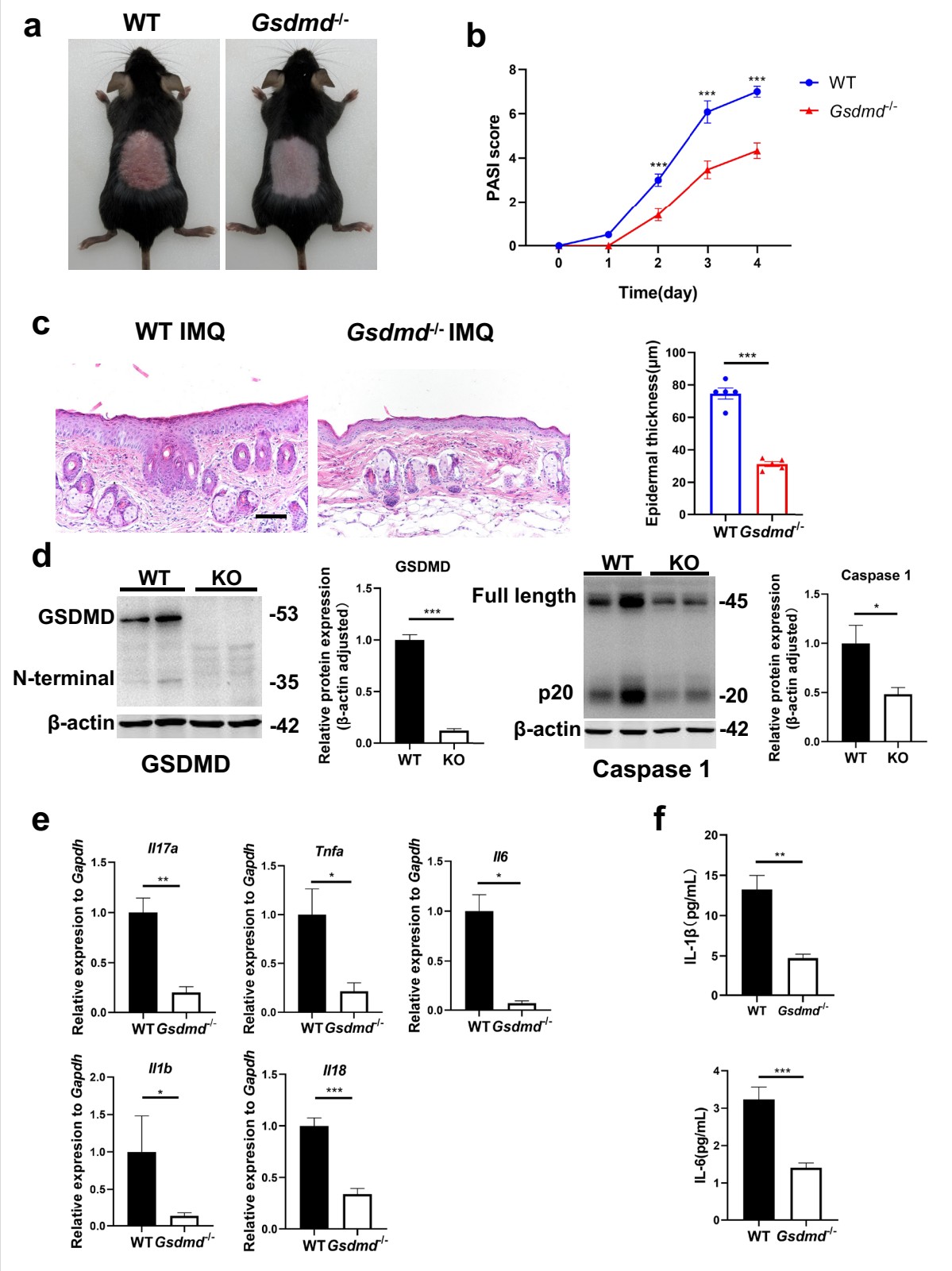

**Figure 2.** GSDMD deficiency mitigates psoriasis-like inflammation in mice. (**a**) Macroscopic phenotypic representation of the dorsal skin in WT and GSDMD-KO mice treated with IMQ at day 4. (**b**) Disease severity of psoriasis induced by IMQ in mice as assessed by PASI score (n=10–12). (**c**) Representative images and statistical analysis of hematoxylin and eosin staining of the dorsal skin in WT and GSDMD-KO mice treated with IMQ at day 4 (n=5). Scale bar = 100 μm. (**d**) Representative images and statistical analysis of western blot analysis showing the expression level of GSDMD and

*Figure 2 continued on next page*

*Figure 2 continued*

caspase 1 in dorsal skin of WT and GSDMD-KO mice treated with IMQ at day 4 (n=4). (**e**) Quantitative PCR analysis of the relative mRNA expression of proinflammatory cytokines in the dorsal skin of WT and GSDMD-KO mice treated with IMQ at day 4 (n=4). Data were normalized to a reference gene, GAPDH. (**f**) ELISA analysis of IL-6 and IL-1β per 1 mg of the dorsal skin from WT and GSDMD-KO mice treated with IMQ at day 4 (n=5). ELISA, enzyme-linked immunosorbent assay; IMQ, imiquimod; WT, wild-type; KO, knockout; PASI, psoriasis area and severity index. Error bars show mean ± SEM. *p<0.05, **p<0.01, ***p<0.001. Data are representative of three independent experiments for (**a, c, d**).

The online version of this article includes the following source data for figure 2:

**Source data 1.** PDF file containing original western blots for *Figure 2d*, indicating the relevant bands and treatments.

**Source data 2.** Original files for western blot analysis displayed in *Figure 2d*.

effect of GSDMD in cKO mice on other cells is quite small. We isolated bone marrow-derived neutrophils and demonstrated by immunoblotting that GSDMD expression was reduced in cKO neutrophil (*Figure 4a*). Then, we applied IMQ to the cKO and control mice (Gsdmd$^{fl/fl}$; S100a8$^{+/+}$) mice on a continuous basis for 4 days to induce psoriasis. Our results showed that after induction of psoriasis, cKO mice had milder disease than controls, as evidenced by reduced PASI scores and the apparent remission of skin changes shown by HE staining, including acanthosis, parakeratosis, and infiltration of leukocytes (*Figure 4b–d*). The lower mRNA expression levels of inflammatory cytokines, including *Il17a, Tnfa, Il6, Il1b,* and *Il18,* were shown in cKO mice, and protein expression of IL-1β and IL-6 also significantly decreased in cKO mice (*Figure 4e–f*). Notably, although neutrophil-specific deletion of GSDMD mice showed a reduction in psoriasis-like symptoms after induction, the reduction was not as pronounced as in GSDMD full knockout mice, suggesting that additional cellular pyroptosis may also be involved in psoriasis pathogenesis. To further investigate neutrophilic pyroptosis in psoriasis, we detected the expression of GSDMD and its N-terminal fragments by WB in the psoriatic skin of WT mice, Gsdmd$^{-/-}$ mice, and cKO mice. We found that after psoriasis induction, the GSDMD N-terminus was only weakly expressed in the skin of cKO mice, whereas it was strongly expressed in WT mice (*Figure 4g*), suggesting that GSDMD-mediated neutrophil pyroptosis is one of the important components of cell pyroptosis in psoriasis. In addition, we used Ly6G antibody to eliminate neutrophils in cKO mice and control mice (*Figure 4h*). It was found that the difference in lesions between the two groups was abolished after neutrophil depletion (*Figure 4i*, *Figure 4—figure supplement 1a*), indicating that GSDMD in neutrophil plays an important role in the pathogenesis of IMQ-induced psoriasis-like lesions in mice.

Considering that pyroptosis is a form of regulated cell death that results in inflammation, its effects can be linked to the release of inflammatory cytokines through membrane pores formed by GSDMD. While neutrophils do undergo pyroptosis, they may exhibit resistance to this cytolytic effect, leading to sustained cytokine release. IL-1β, a significant inflammatory cytokine of pyroptosis, has a crucial function in the pathogenesis of psoriasis. The IL-1β-IL-1R pathway promotes the ailment by regulating IL-17-producing cells in the dermis and stimulating keratinocytes to magnify inflammatory cascades. Therefore, it is hypothesized that the occurrence of neutrophil pyroptosis in psoriasis may activate inflammatory pathways downstream by releasing cytokine IL-1β. In order to further verify our hypothesis, we analyzed publicly available single-cell sequencing data. We found that in the skin of psoriasis-like mice, *Il1b* and *Il6* were mainly expressed in neutrophils, while *Il17a* was mainly expressed in T cells, *Il18* was mainly expressed in macrophages and fibroblasts, and *Tnfa* was mainly expressed in macrophages and T cells (*Figure 4—figure supplement 1b*). This suggests that neutrophil pyroptosis may affect the secretion of cytokines such as IL-1β and IL-6 by neutrophils, thereby affecting the functions of other immune cells such as T cells and macrophages, forming a complex inflammatory network in psoriasis and participating in the pathogenesis of psoriasis.

## Discussion

In this study, we dissected the role of GSDMD protein, a key pyroptosis executioner, in the context of psoriasis. We found that GSDMD protein was activated in the skin of psoriasis, and that GSDMD in neutrophils promotes psoriasis inflammation through pyroptosis and the release of cytokines. Research on pyroptosis has tended to focus on infections, tumors, and other acute inflammations. Pyroptosis is not only involved in the regulation of sepsis, but also mediates sepsis-related organ damage (*Zheng et al., 2021*). This means that while moderate pyroptosis promotes clearance of pathogens, excessive

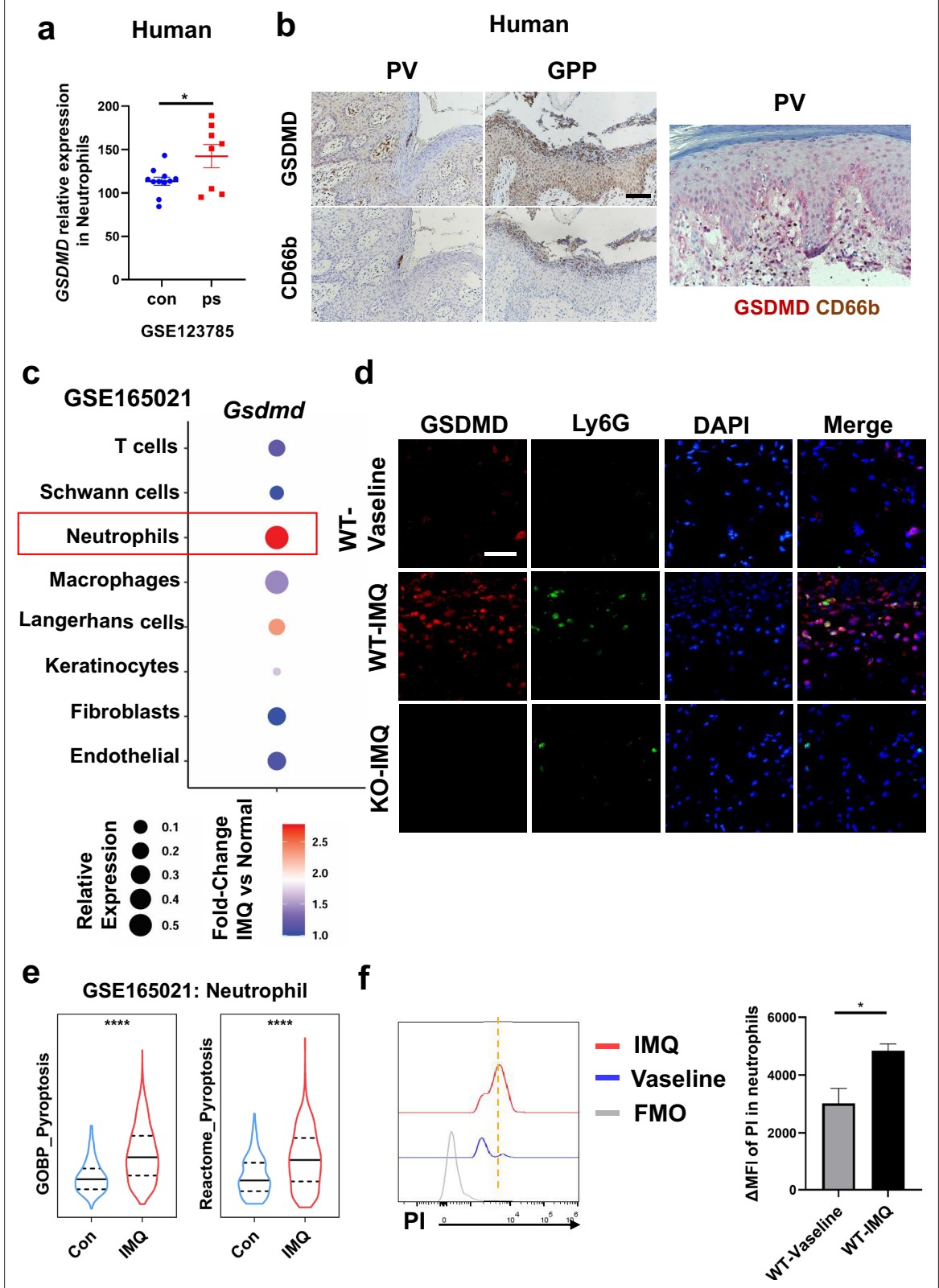

**Figure 3.** Neutrophils undergo GSDMD-mediated pyroptosis in psoriasis. (**a**) Expression of *GSDMD* in neutrophils in psoriasis patients and healthy people from GSE123785. (**b**) Representative immunohistochemical staining of CD66b and GSDMD in two consecutive sections of skin tissue from patients with psoriasis vulgaris or generalized pustular psoriasis (left); representative immunohistochemical staining of CD66b (brown) and GSDMD (red) in sections of skin tissue from patients with psoriasis vulgaris (n=3–5, each group). Scale bar = 100 µm. (**c**) Dot plot of *Gsdmd* expression in each cell

*Figure 3 continued on next page*

*Figure 3 continued*

type in the skin of IMQ-induced psoriasis-like mice (GSE165021). (**d**) Representative immunofluorescence of GSDMD (red), Ly6G (green), and nuclear (blue) in dorsal skin of WT mice treated with vehicle, WT mice treated with IMQ, and GSDMD-KO mice treated with IMQ at day 5. Scale bar = 50 μm. (**e**) ssGSEA showed the expression of pyroptosis-related genes in neutrophils infiltrating in control and IMQ-induced psoriasis-like tissue (GSE165021). (**f**) Representative and statistical graphs of the mean fluorescence intensity of propidium iodide in neutrophils from skin of WT mice as detected by flow cytometry (n=3). ps, psoriasis skin; con, control; PV, psoriasis vulgaris; GPP, generalized pustular psoriasis; IMQ, imiquimod; WT, wild-type; MFI, mean fluorescence intensity; PI, propidium iodide; MFO, fluorescence minus one. Error bars show mean ± SEM. *p<0.05, ****p<0.0001. Data are representative of three independent experiments for (**b, d–e**).

The online version of this article includes the following figure supplement(s) for figure 3:

**Figure supplement 1.** The expression of *Ly6g* and *S100a8* in the skin of IMQ-induced psoriasis-like mice, and TUNEL assay in neutrophils.

pyroptosis can lead to uncontrolled inflammatory responses and promote the onset and development of sepsis (*Zheng et al., 2021*; *Aglietti and Dueber, 2017*; *Shi et al., 2015*). Moreover, pyroptosis promotes severe pancreatitis and associated lung injury through the release of cytokines IL-1β and IL-18 (*Wu et al., 2021*), and is implicated in acute myocardial injury and spinal cord injury (*Shi et al., 2021*; *Al Mamun et al., 2021*). In addition, GSDMD-induced pyroptosis of antigen-presenting cells in tumor microenvironments impairs antigen presentation in response to anti-PD-L1 treatment (*Jiang et al., 2022*). Only in recent years, studies have explored the function of pyroptosis in psoriasis. Iwona Flisiak et al. found that GSDMD expression was significantly increased in the skin of psoriasis patients, suggesting that it may be involved in the pathogenesis of psoriasis (*Nowowiejska et al., 2023*). Xu Chen et al. reported that GSDMD-mediated keratinocyte pyroptosis promotes excessive proliferation and abnormal differentiation induced by the immune microenvironment in psoriatic skin inflammation, thus contributing to the pathogenesis of psoriasis (*Lian et al., 2023*). In addition, there are some studies on pyroptosis-related proteins upstream of GSDMD, such as NLPR3 inflammasome and AIM2 inflammasome, which are involved in and promote psoriasis through downstream inflammatory cytokines (*Deng et al., 2019*; *Verma et al., 2021*; *Zhang et al., 2023*).

Consistent with earlier research, our study found that the expression of GSDMD and its upstream molecule caspase 1 was upregulated in the skin of psoriasis patients and psoriasis-like mice, indicating that the pyroptosis pathway is activated in psoriasis skin lesions. Knockout of GSDMD in vivo is effective in significantly ameliorating IMQ-induced psoriasis-like lesions and reducing the level of skin inflammation, which also notably reduces the expression of the pyroptosis-inducing inflammatory cytokine IL-1β. In our study, we found that the upregulated GSDMD signal was more significant in neutrophils in both human and animal tissues. In addition, specific knockout of GSDMD in neutrophils in vivo can also yield results consistent with GSDMD KO mice, although not as obvious as in KO mice. Our study is for the first time demonstrating that GSDMD-mediated neutrophil pyroptosis in psoriasis is involved in psoriatic inflammation and may affect the inflammatory network in psoriasis. Considering the role and early recruitment of neutrophils in psoriasis (*Chiang et al., 2019*; *Wang and Jin, 2020*), it is readily accepted that neutrophils act as innate immune cells in response to changes in the microenvironment to initiate pyroptosis signaling and release cytokines to engage and maintain psoriasis inflammation. IL-1β released by pyroptosis may amplify the inflammatory cascade through the IL-1β-IL-1R pathway via direct regulation of dermal IL-17-producing cells and stimulation of keratinocytes (*Cai et al., 2019*), thereby contributing to skin inflammation and psoriasis. In addition, the unique resistance of neutrophils to pyroptosis (*Dąbrowska et al., 2019*) makes it reasonable for them to continuously release cytokines after activation of the pyroptosis pathway and further stimulate skin inflammation.

Our research primarily focuses on the role of neutrophil pyroptosis in psoriasis, this does not conflict with existing reports indicating that KC cell pyroptosis also contributes to disease progression (*Lian et al., 2023*). Both studies underscore the significant role of GSDMD-mediated pyroptotic signaling in psoriasis, and the consistent involvement of KC cells and neutrophils further emphasizes the potential therapeutic value of targeting GSDMD signaling in psoriasis treatment. Nevertheless, combining previous studies and our findings, we can believe that GSDMD-mediated pyroptosis, which is a dynamic biological process, particularly neutrophil pyroptosis, is involved in the inflammation of psoriasis, especially in the early stage. This provides a potential basis for the possible use of drugs targeting pyroptosis, especially drugs targeting neutrophil pyroptosis, in the treatment of psoriasis. Dimethyl fumarate is a type of medicine widely used to treat psoriasis in Europe (*Mrowietz et al.,*

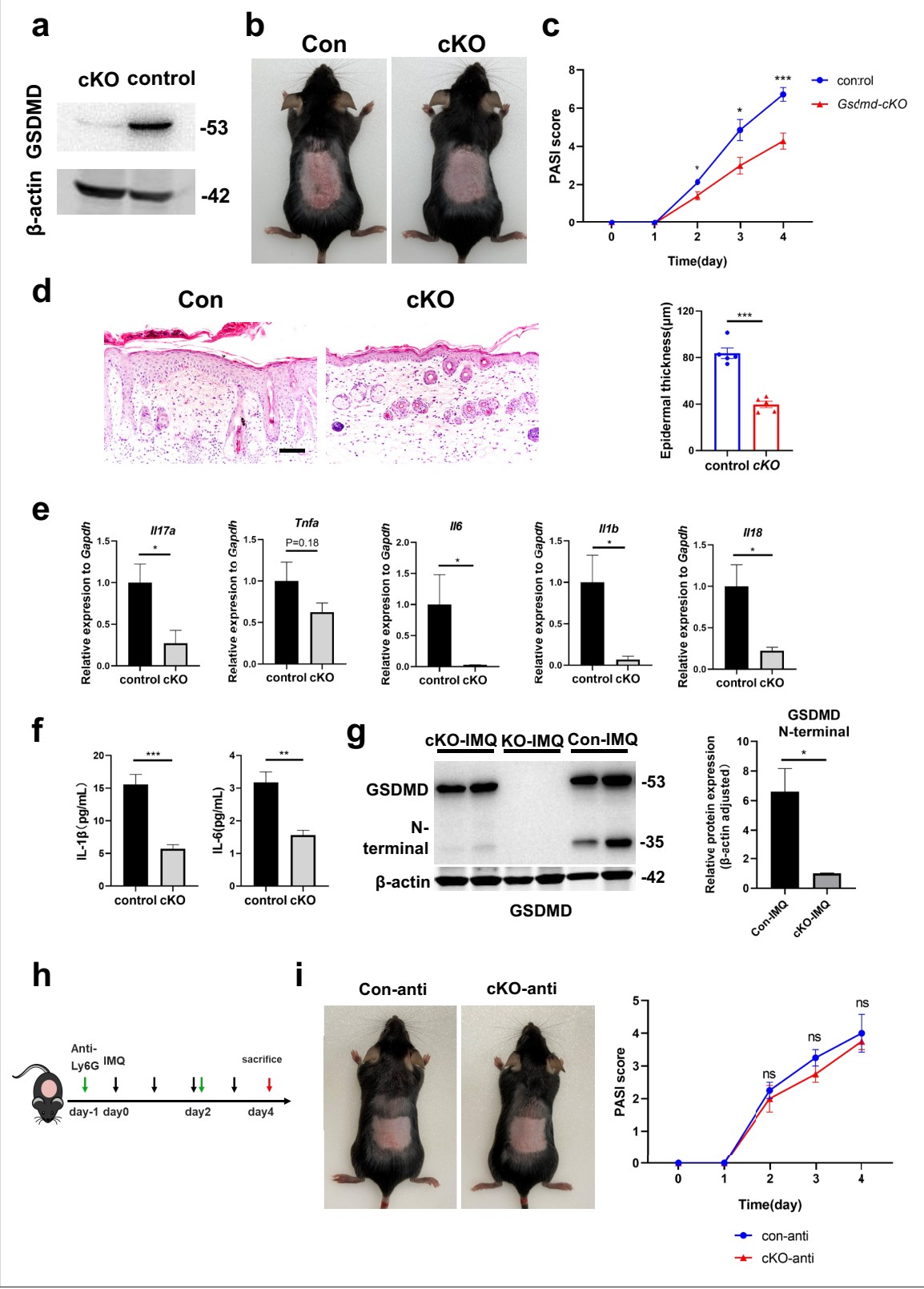

**Figure 4.** GSDMD depletion in neutrophils attenuates the development of skin inflammation in psoriasis. (**a**) Representative images of western blot showing the expression of GSDMD in bone marrow-derived neutrophils of cKO and isotype control mice. (**b**) Macroscopic phenotypic representation of the dorsal skin in control and GSDMD-cKO mice treated with IMQ at day 4. (**c**) Disease severity of psoriasis induced by IMQ in mice as assessed by PASI score (n=7). (**d**) Representative images and statistical analysis of hematoxylin and eosin staining of the dorsal skin in control and GSDMD-cKO

*Figure 4 continued on next page*

*Figure 4 continued*

mice treated with IMQ at day 4 (n=5). Scale bar = 100 μm. (**e**) Quantitative PCR analysis of the relative mRNA expression of proinflammatory cytokines in the dorsal skin of control and GSDMD-cKO mice treated with IMQ at day 4 (n=4). Data were normalized to a reference gene, GAPDH. (**f**) ELISA analysis of IL-6 and IL-1β per 1 mg of the dorsal skin from control and GSDMD-cKO mice treated with IMQ at day 4 (n=5). (**g**) Representative images and statistical analysis of western blot analysis showing the expression level of GSDMD and its N-terminal fragments in dorsal skin of WT, cKO, and *Gsdmd*[-/-] mice treated with IMQ at day 4 (n=4). (**h**) Schematic representation of the use of anti-Ly6G antibody in the IMQ-induced psoriasis mouse model. (**i**) Macroscopic phenotypic representation and PASI score of cKO mice and control mice treated with IMQ and Ly6G antibody. cKO, conditional knockout; ELISA, enzyme-linked immunosorbent assay; IMQ, imiquimod; PASI, psoriasis area and severity index, WT, wild-type. Error bars show mean ± SEM. *p<0.05, **p<0.01, ***p<0.001. Data are representative of three independent experiments for (**a, b, d, g, i**).

The online version of this article includes the following source data and figure supplement(s) for figure 4:

**Source data 1.** PDF file containing original western blots for *Figure 4a and g*, indicating the relevant bands and treatments.

**Source data 2.** Original files for western blot analysis displayed in *Figure 4a and g*.

**Figure supplement 1.** Flow cytometry plots of neutrophil, and data analysis in single-cell RNA sequencing from the skin of IMQ-induced psoriasis-like mice.

*2018*; *Mrowietz and Asadullah, 2005*). The latest research proves that dimethyl fumarate can inhibit the function of GSDMD and thereby inhibit pyroptosis-related inflammation (*Humphries et al., 2020*).

There is some limitation in our study such as the lack of exploration of upstream signaling involved in neutrophil pyroptosis along with its specific molecular mechanisms. Although we observed significantly increased GSDMD in neutrophils in pustular psoriasis, we were constrained to studying the established PV animal model due to the current absence of a mature GPP animal model. This represents a limitation of our study. Nevertheless, our study showed that neutrophil pyroptosis is involved in and contributes to psoriasis inflammation, indicating potential avenues and foundations for novel psoriasis treatment strategies.

## Methods
### Experimental animals
All mice were of the species *Mus musculus* (C57BL/6). All experiments were performed with female mice (6–8 weeks of age). The *Gsdmd*[-/-] mice were provided by Dr. Feng Shao (National Institute of Biological Sciences, Beijing, China). The *Gsdmd*[fl/fl] mice were generated using conditional gene targeting methods as described previously (*Li et al., 2019*). The *S100a8*-Cre-EGFP transgenic mouse line was originally created in Dr. Irving L Weissman's laboratory before being deposited in The Jackson Laboratory (stock #021614; Bar Harbor, ME, USA). All mice were kept in a barrier facility, and all animal experiments were conducted in accordance with the procedure approved by the Ethical Review Committee for Laboratory Animal Welfare of the Nanjing Medical University (IACUC-2203023).

### Psoriasis-like mouse model
Psoriasis-like mice were induced by 5% IMQ cream as previously described (*Kim et al., 2023*). Mice received topical application of 32.5 mg 5% IMQ to the shaved dorsal skin of a limited area (2*3 cm²) for 4 days, and control mice were given the same dose of Vaseline for 4 days. The skin was examined every day, and the dorsa were photographed. The mice were sacrificed by cervical dislocation on day 5, and their skin and serum were collected for the next experiment.

### Western blot
Mice back skin tissues were lysed in lysis buffer solution (150 mM NaCl, 10 mM Tris [pH 7.4], 5 mM EDTA, 1 mM EGTA, and 0.1% NP-40) supplemented with 1 mM phenylmethylsulphonyl fluoride, and complete protease inhibitor 'cocktail' (Sigma-Aldrich), followed by tissue homogenization and incubation for 60 min at 4°C. The lysates were centrifuged for 10 min at 14,000×*g*, and supernatants were denatured with SDS buffer, and boiled for 10 min. Proteins were separated by SDS-polyacrylamide gel electrophoresis and transferred onto nitrocellulose membranes. The membranes were immunoblotted with primary antibodies and proteins detected with appropriate secondary anti-rabbit antibody conjugated to fluorescence. Immunoreactivity was visualized by the Odyssey Imaging System (LI-COR Biosciences).

## Enzyme-linked immunosorbent assay

Homogenized the mouse back skin tissues (the method is the same as before), and collected the supernatant. Supernatants were collected and measured for the level of IL-1β (DY401) and IL-6 (DY406) according to the manufacturer's instructions by sandwich ELISA (R&D Systems).

## Immunofluorescence and immunohistochemistry staining

For immunofluorescence, the mice skin tissues were collected, fixed in 4% buffered formaldehyde, and embedded in paraffin. Tissue sections were incubated at 4°C overnight with primary antibody to GSDMD (Abcam, ab219800, 1:300) and Ly6G (BDscience, Cat551459, 1:100). Slides were then incubated with indicated secondary antibodies. TUNEL staining was carried out with staining kit (Servicebio). The nuclei were counterstained with 4',6-diamidino-2-phenylindole (Sigma-Aldrich). Slides were dried and mounted using ProLong Antifade mounting medium (Beyotime Biotechnology). Slides were visualized using a Nikon 50i fluorescence microscope. For immunohistochemical staining, human skin tissue samples were obtained from the Department of Dermatology, Jiangsu Provincial People's Hospital. Our study was approved by the Ethics Committee of the First Affiliated Hospital of Nanjing Medical University (Jiangsu Province Hospital), Ethics Number: 2020-SRFA-101, and informed consent was obtained from all study participants. Sections were blocked and incubated with primary antibodies to GSDMD (Abcam, ab215203, 1:1000) and CD66b (BioLegend, Cat305102, 1:80) after heat-induced antigen retrieval. Slides were then incubated with horseradish peroxidase-conjugated secondary antibodies. Diaminobenzidine was used for detection. Images were captured with a Nikon 50i microscope. Images were processed using ImageJ 1.53c and AdobePhotoshopCS6 software.

## FCM and FACS

For FCM, skin tissue was digested into single-cell suspensions with collagenase P (ROCHE, 1 mg/mL), DNaseI (ROCHE, 100 µg/mL), and HAase (ABSIN, 10 µg/mL) at 37°C for 1 hr. Single-cell suspensions were stained with anti-CD45-Alexa Flour 700 (eBioscience, 1:400), anti-Ly6G-eFlour 450 (eBioscience, 1:400), and FVD eFlour 506 (eBioscience, 1:1000) for FCM (Thermo). After the mice were euthanized, the long bones were separated and the bone marrow was aspirated with a sterile syringe. Single-cell suspension was stained with anti-CD45-Alexa Flour 700 (eBioscience, 1:400), anti-CD11b-Percp-cy5.5 (eBioscience, 1:400), and anti-Ly6G-eFlour 450 (eBioscience, 1:400) for fluorescence-activated cell sorting (FACS) analysis (Thermo), $CD45^+FVD^-CD11b^+Ly6G^+$ neutrophils were obtained by FACS. All FCM analyses were performed on an Attune NxT Flow Cytometer (Thermo Fisher Scientific), and data were analyzed using FlowJo 10 software.

## Neutrophil depletion

An injection of anti-Ly6G antibody (Selleck, 200 µg) was given on day 1 and day 2 during continuous IMQ application to deplete neutrophil in mice.

## Statistical analysis

Data were analyzed using Prism 8 (GraphPad, San Diego, CA, USA). Comparisons between two groups were analyzed by using unpaired Student's t-test. All data are presented as the mean ± SEM and are representative of at least three independent experiments. $p < 0.05$ was considered indicative of statistical significance. *$p \leq 0.05$, **$p < 0.01$, ***$p < 0.001$, ****$p \leq 0.0001$, ns $p > 0.05$.

## Additional information

### Funding

| Funder | Grant reference number | Author |
| --- | --- | --- |
| National Natural Science Foundation of China | 82073439 | ZhiQiang Yin |
| National Natural Science Foundation of China | 82373475 | ZhiQiang Yin |

| Funder | Grant reference number | Author |
| --- | --- | --- |

The funders had no role in study design, data collection and interpretation, or the decision to submit the work for publication.

## Author contributions

Jian Liu, Data curation, Investigation, Methodology, Writing - original draft; YuYing Jiang, ZiYue Diao, Data curation, Investigation, Methodology; DanDan Chen, RuiYuan Xia, Methodology; BingWei Wang, Shuo Yang, ZhiQiang Yin, Conceptualization, Supervision, Project administration, Writing – review and editing

## Author ORCIDs

YuYing Jiang ⓘD https://orcid.org/0000-0002-3263-7730
ZhiQiang Yin ⓘD https://orcid.org/0000-0002-6510-1051

## Ethics

Human skin tissue samples were obtained from the Department of Dermatology, First Affiliated Hospital of Nanjing Medical University, which was approved by the Ethics Committee of the First Affiliated Hospital of Nanjing Medical University (Jiangsu Province Hospital), Ethics Number: 2020-SRFA-101, and informed consents were obtained from all study participants.

All mice were kept in a barrier facility, and all animal experiments were conducted in accordance with the procedure approved by the Ethical Review Committee for Laboratory Animal Welfare of the Nanjing Medical University (IACUC-2203023).

Reviewer #1 (Public review): https://doi.org/10.7554/eLife.101248.3.sa1
Reviewer #2 (Public review): https://doi.org/10.7554/eLife.101248.3.sa2
Author response https://doi.org/10.7554/eLife.101248.3.sa3

# Additional files

## Supplementary files

• MDAR checklist

## Data availability

All data generated or analysed during this study are included in the manuscript and supporting files; source data files have been provided for Figures 1, 2 and 4.

The following previously published datasets were used:

| Author(s) | Year | Dataset title | Dataset URL | Database and Identifier |
| --- | --- | --- | --- | --- |
| Garcia-Rubio R, Jimenez-Ortigosa C, DeGregorio L, Quinteros C | 2021 | Multifactorial Role of Mitochondria in Echinocandin Tolerance Revealed by Transcriptome Analysis of Drug-Tolerant Cells | https://www.ncbi.nlm.nih.gov/geo/query/acc.cgi?acc=GSE178797 | NCBI Gene Expression Omnibus, GSE178797 |
| Choy DF, Hsu DK, Seshasayee D, Fung MA | 2020 | Genome-Wide Profiling of Lesional and Non-Lesional Skin from Atopic Dermatitis, Psoriasis, and Contact Dermatitis Skin | https://www.ncbi.nlm.nih.gov/geo/query/acc.cgi?acc=GSE153007 | NCBI Gene Expression Omnibus, GSE153007 |
| Ronholt K, Langkilde A, Johansen C, Vestergaard C, Fauerbye A, López-Vales R, Dinarello C, Iversen L | 2022 | Expression data from lesional (LP) and non-lesional (NP) skin in patients with psoriasis | https://www.ncbi.nlm.nih.gov/geo/query/acc.cgi?acc=GSE161683 | NCBI Gene Expression Omnibus, GSE161683 |

*Continued*

| Author(s) | Year | Dataset title | Dataset URL | Database and Identifier |
|---|---|---|---|---|
| Sun Y, Chen D, Zhu Y, Wu Z | 2021 | Single-cell transcriptomics of the mouse skin reveal potential target of psoriasis | https://www.ncbi.nlm.nih.gov/geo/query/acc.cgi?acc=GSE165021 | NCBI Gene Expression Omnibus, GSE165021 |
| Catapano M, Vergnano M, Romano M, Mahil SK | 2019 | Neutrophils RNAseq from Generalised Pustular Psoriasis patients and healthy individuals | https://www.ncbi.nlm.nih.gov/geo/query/acc.cgi?acc=GSE123785 | NCBI Gene Expression Omnibus, GSE123785 |

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
