## [Editor Report · eLife Assessment]

This is a **valuable** study regarding the role of gasdesmin D in experimental psoriasis. The study contains **solid** evidence for such a role, involving neutrophils, from murine models of skin inflammation, as well as correlative data of elevated gasdermin D expression in human psoriatic skin. The findings will be of interest to researchers trying to unravel pathways of skin inflammation.

---

## [Referee Report · Reviewer #1 (Public review)]

Summary:

Recommendations for the authors In this study, Liu, Jiang, Diao et.al. investigated the role of GSDMD in psoriasis-like skin inflammation in mice. The authors have used full-body GSDMD knock-out mice and Gsdm floxed mice crossed with the S100A8- Cre. In both mice, the deficiency of GSDMD ameliorated the skin phenotype induced by the imiquimod. The authors also analyzed RNA sequencing data from the psoriatic patients to show an elevated expression of GSDMD in the psoriatic skin.

Strengths:

It has the potential to unravel the new role of neutrophils.

Comments on revisions:

The authors have addressed the majority of comments and concerns and highlighted the potential limitations wherever not possible.

---

## [Referee Report · Reviewer #2 (Public review)]

Summary:

The authors describe elevated GSDMD expression in psoriatic skin, and knock-out of GSDMD abrogates psoriasis-like inflammation.

Strengths:

The study is well conducted with transgenic mouse models. Using mouse-models with GSDMD knock-out showing abrogating inflammation, as well as GSDMD fl/fl mice without neutrophils having a reduced phenotype.

My major concern would be the involvement of other inflammasome and GSDMD bearing cell types, esp. Keratinocytes (KC), which could be an explanation why the experiments in Fig 4 still show inflammation.

Comments on revisions:

The authors have sufficiently addressed my questions.

---

## [Author Response]

The following is the authors’ response to the original reviews.

**eLife Assessment**
This is a potentially interesting study regarding the role of gasdesmin D in experimental psoriasis. The study contains useful data from murine models of skin inflammation, however the main claims (on neutrophil pyroptosis) are incompletely supported in its current form and require additional experimental support to justify the conclusions made.

We sincerely appreciate the positive assessment regarding the significance of our study, as well as the valuable suggestions provided by the reviewers. We have included new data, further discussions and clarifications in the revised manuscript to adequately address all the concerns raised by the reviewers and better support our conclusions.

**Public Reviews:**

**Reviewer #1 (Public review):**
Summary:In this study, Liu, Jiang, Diao et.al. investigated the role of GSDMD in psoriasis-like skin inflammation in mice. The authors have used full-body GSDMD knock-out mice and Gsdm floxed mice crossed with the S100A8- Cre. In both mice, the deficiency of GSDMD ameliorated the skin phenotype induced by the imiquimod. The authors also analyzed RNA sequencing data from the psoriatic patients to show an elevated expression of GSDMD in the psoriatic skin.Overall, this is a potentially interesting study, however, the manuscript in its current format is not completely a novel study.Strengths:It has the potential to unravel the new role of neutrophils.Weaknesses:The main claims are only partially supported and have scope to improve

We thank the reviewer for the positive evaluation of the interest and potential of our work. In response to reviewers’ suggestions, we have added new content, including additional data and discussions, to further demonstrate the important role of GSDMD-mediated neutrophil pyroptosis in the pathogenesis of psoriasis, thereby enhancing the completeness of our research.

**Reviewer #2 (Public review):**
Summary:The authors describe elevated GSDMD expression in psoriatic skin, and knock-out of GSDMD abrogates psoriasis-like inflammation.Strengths:The study is well conducted with transgenic mouse models. Using mouse-models with GSDMD knock-out showing abrogating inflammation, as well as GSDMD fl/fl mice without neutrophils having a reduced phenotype.I fear that some of the conclusions cannot be drawn by the suggested experiments. My major concern would be the involvement of other inflammasome and GSDMD bearing cell types, esp. Keratinocytes (KC), which could be an explanation why the experiments in Fig 4 still show inflammation.Weaknesses:The experiments do not entirely support the conclusions towards neutrophils.

We appreciate the reviewers’ positive evaluation regarding the application of our mouse models. We also thank the reviewers for insightful comments and suggestions that can improve the quality of our work. Addressing these issues has significantly strengthened our conclusions. Our responses to the above questions are as follows.

Specific questions/comments:Fig 1b: mainly in KC and Neutrophils?

In Figure 1b, we observed that GSDMD expression is higher in the psoriasis patient tissues compared to control samples. As the role of GSDMD in keratinocytes during the pathogenesis of psoriasis has already been explored[1], we focused our study on GSDMD in neutrophils. In response to the comments, we have added co-staining results of the neutrophil marker CD66b and GSDMD in the revised manuscript (see new Figure 3b in the revised manuscript). This addition further substantiates the expression of GSDMD in neutrophils within psoriasis tissue.

Fig 2a: PASI includes erythema, scaling, thickness and area. Guess area could be trick, esp. in an artificial induced IMQ model (WT) vs. the knock-out mice.

In our model, to accurately assess the disease condition in mice, we standardized the drug treatment area on the dorsal side (2*3 cm). Therefore, the area was not factored into the scoring process, and we have included a detailed description of this in the revised manuscript.

Fig 2d: interesting finding. I thought that CASP-1 is cleaving GSDMD. Why would it be downregulated?

Regarding the downregulation of CASP in GSDMD KO mouse skin tissue, existing studies indicate that GSDMD generates a feed-forward amplification cascade via the mitochondria-STING-Caspase axis [2]. We hypothesize that the absence of GSDMD attenuates STING signaling’s activation of Caspase.

Line 313: as mentioned before (see Fig 1b). KC also show a stron GSDMD staining positivity and are known producers of IL-1b and inflammasome activation. Guess here the relevance of KC in the whole model needs to be evaluated.

Our research primarily focuses on the role of neutrophil pyroptosis in psoriasis, this does not conflict with existing reports indicating that KC cell pyroptosis also contributes to disease progression[1]. Both studies underscore the significant role of GSDMD-mediated pyroptotic signaling in psoriasis, and the consistent involvement of KC cells and neutrophils further emphasizes the potential therapeutic value of targeting GSDMD signaling in psoriasis treatment. We have expanded upon this discussion in the revised manuscript.

Fig 4i - guess here the conclusion would be that neutrophils are important for the pathogenesis in the IMQ model, which is true. This experiment does not support that this is done by pyroptosis.

To address the question, we analyzed the publicly available single-cell transcriptomic data (GSE165021) and found that, compared to the control group, neutrophils infiltrating in IMQ-induced psoriasis-like tissue display a higher expression of pyroptosis-related genes (see new Figure 3e in the revised manuscript). These results strengthen our conclusions about the role of neutrophil pyroptosis in the progression of psoriasis.

**Recommendations for the authors:**

**Reviewer #1 (Recommendations for the authors):**
Specific Comments:• Figure 1: Micro abscesses would already be dead, which would likely reflect as non-specific staining. Authors should consider double staining (e.g., GSDMD+Ly6G).

We thank the reviewer for the useful suggestion. We have added co-staining results of the neutrophil marker CD66b and GSDMD in the revised manuscript (see new Figure 3b in the revised manuscript). This addition further substantiates the expression of GSDMD in neutrophils within psoriasis tissue.

• Figures 1 b, c, and d do not have the n number for representative experiments and images.

We apologize for our oversight. We have added the relevant information in the revised manuscript and have reviewed and corrected the entire text.

• What is the difference between psoriasis patients in Figure 1 versus Figure 3 as the staining patterns are different? It is difficult to interpret from Figure 1 that expression is limited to neutrophils. Authors should consider double staining (e.g., GSDMD+Ly6G). How many samples were stained to draw this conclusion?

We thank the reviewer for the suggestion. In Figure 1b, we observed that GSDMD expression is higher in the psoriasis patient tissues compared to control samples. We have added co-staining results of the neutrophil marker CD66b and GSDMD in the revised manuscript (see new Figure 3b in the revised manuscript). For each staining group, we examined samples from 3-5 patients to draw the conclusion.

• Figure 2: GSDMD deficiency mitigates psoriasis-like inflammation in mice has been shown before (PMID#37673869). The paper showed that the GSDMD was mainly expressed in keratinocytes. What is the view of the authors on it and how does this data correlate with the data presented in this manuscript by the authors?

Consistent with previous studies[1], we observed increased expression of pyroptosis-related proteins in psoriatic lesions. However, our research focused specifically on the role of neutrophil pyroptosis in psoriasis, this does not conflict with existing reports indicating that KC cell pyroptosis also contributes to disease progression. Both studies underscore the significant role of GSDMD-mediated pyroptotic signaling in psoriasis, and the consistent involvement of KC cells and neutrophils further emphasizes the potential therapeutic value of targeting GSDMD signaling in psoriasis treatment. We have expanded upon this discussion in the revised manuscript.

• Figure 3d: It is unclear if the IF shows an epidermal or dermal area. As shown by authors in other figures (human psoriatic skin), do authors observe more GSDMD in the micro abscess, which is localized in the epidermis? The authors should also show the staining of GSDM/Ly6G in the whole skin sample.

The region we presented for immunofluorescence staining corresponds to the dermis of the mice, as we did not observe typical neutrophil micro abscesses similar to those in human psoriasis in the epidermis of IMQ-induced classical psoriasis vulgaris (PV) model. Therefore, we have only shown the staining in the dermal area.

• Figure 3e: PI staining also represents necrotic cells and TUNEL staining would not represent just apoptotic cells. It is unclear how the authors conclude an ongoing pyroptosis in neutrophils. A robust dataset is needed to provide evidence supporting neutrophil pyroptosis in the IMQ-challenged mice.

We thank the reviewer for the valuable suggestion. GSDMD is the effector protein of pyroptosis. To further confirm that cells are undergoing pyroptosis, it is necessary to morphologically stain the GSDMD N-terminal protein. Although there is currently no GSDMD N-terminal fluorescent antibody available, we detected the cleaved N-terminus of GSDMD by WB in mouse psoriasis-like skin tissue, and its increased expression suggested increased cell pyroptosis (see new Figure 1d in the revised manuscript). Moreover, we analyzed the publicly available single-cell transcriptomic data (GSE165021) and found that, compared to the control group, neutrophils infiltrating in IMQ-induced psoriasis-like tissue display a higher expression of pyroptosis-related genes (see new Figure 3e in the revised manuscript). These results strengthen our conclusions about the role of neutrophil pyroptosis in the progression of psoriasis.

• Figure 4: The authors did not clarify the reason for choosing D4 over the usual D7 for the imiquimod experiment. S100A8-Cre is also reported in monocytes and granulocytes/monocyte progenitors. And, the authors also show the expression in macrophages and neutrophils, but in the text, only neutrophils are mentioned. The authors should state the results in the text as well to avoid misrepresentation of the data.

We thank the reviewer for the useful suggestion. We have repeated many times of experiments in our previous studies and observed that the IMQ-induced mouse psoriasis model showed the obvious signs of self-resolution after Day 4 even with continuing topical IMQ application, thus we chose 4 days over 7 days for the imiquimod experiment, which are consistent with many other studies[3, 4].

Many studies use S100A8-Cre mice for neutrophil-specific gene knockout[5, 6]. Moreover, we used Ly6G antibody to eliminate neutrophils in GSDMD-cKO mice and control mice. It was found that the difference in lesions between the two groups was abolished after neutrophil depletion, indicating that neutrophil pyroptosis plays an important role in the pathogenesis of imiquimod-induced psoriasis-like lesions in mice. As the database analysis results showed that macrophages have slight expression of *S100a8*, according to the suggestion of the reviewer, we have added a more precise description in the revised manuscript.

• Figure S2a: Ly6G antibody reduced the ly6G positive, but also negative cells compared to PBS. If this is correct, what is the explanation, and how this observation has been considered for concluding results?

Neutrophils play an important role in regulating inflammatory responses, and their deletion can reduce the overall inflammatory level in the body, which also results in a decrease in other non-neutrophil cells. However, this change does not affect our conclusions. Our results show that after the deletion of neutrophils, there is no difference in the pathological manifestations between the cKO group and the control group. This further that GSDMD in neutrophil plays an important role in the pathogenesis of miquimod-induced psoriasis-like lesions in mice.

• The conclusion in Figure 4i is incorrect as Ly6G administration had an effect on the wt, so it shows neutrophils play a role, but not neutrophil pyroptosis.- 321 "It was found that the difference in lesions between the- 321 two groups was abolished after neutrophil depletion (Fig4i, S2a), indicating that- 322 neutrophil pyroptosis plays an important role in the pathogenesis of- 323 imiquimod-induced psoriasis-like lesions in mice"

Our results show that after the deletion of neutrophils, there is no difference in the pathological manifestations between the cKO group and the control group. This further indicates that the lower disease scores observed in cKO mice, in the absence of neutrophil deletion, depend on the presence of neutrophils. In the revised manuscript, we have changed the statement to “It was found that the difference in lesions between the two groups was abolished after neutrophil depletion (Fig4i, S2a), indicating that GSDMD in neutrophil plays an important role in the pathogenesis of miquimod-induced psoriasis-like lesions in mice”

• The effect of LyG Ab: reduced PASI in the wt, but the effect on the ko remains the same. What are the other molecular changes observed? What was the level of neutrophils in the wt and the S1A008Cre GsdmDfl/fl mice under steady state and how are they change upon imiquimod challenge? A complete profiling of the immune cells is needed for all the experiments.

As demonstrated by the results, the deletion of neutrophils did not significantly alter the pathological phenotype of cKO mice. We believe that this outcome precisely highlights the crucial role of GSDMD in regulating neutrophil inflammatory responses.

• Figure S2b: The authors conclude that Il-1b in the imiquimod skin is mainly expressed by neutrophils, but the analysis presented in the figure does not support this conclusion. Both neutrophils and macrophages are majorly positive for I1-b, with some expression on Langerhans and fibroblasts. No n numbers are provided for the experiment

As we discussed in the manuscript, we speculate that neutrophil pyroptosis may release cytokines, which in turn activate other cells to secrete cytokines, forming a complex inflammatory network in psoriasis. This may suggest that neutrophil pyroptosis may be involved in the pathogenesis of psoriasis by affecting the secretion of cytokines such as IL-1B and IL-6 by neutrophils, thereby affecting the function of other immune cells such as T cells and macrophages.

We have added the n number in the revised manuscript.

• For clarity and transparency, a list of antibodies with the associate clone and catalogue number should be provided or integrated into the method text.

We thank the reviewer for the useful suggestion. We have added the associate clone and catalogue number of antibodies used in the method text of revised manuscript.

**Reviewer #2 (Recommendations for the authors):**
Fig 3b: psoriasis and pustular psoriasis have a different pathophysiology (autoimmune vs. autoinflammatory). Neutrophils are centrally important for GPP for the cleavage of IL-36. Guess as not further referred to pustular psoriasis in the paper, that comparison is rather deviating from the story.

In Figure 3b, we stained for GSDMD and CD66b in both plaque psoriasis (PV) and generalized pustular psoriasis (GPP), not to compare the expression differences between the two types of psoriasis, but rather to demonstrate that significant GSDMD expression is present in neutrophils in different types of psoriasis. Unfortunately, due to the lack of a well-established animal model for GPP, we were only able to conduct studies using the established PV animal model. We acknowledge this limitation in our research. In our revised manuscript, we have added the following explanation in the discussion section: “Although we observed significantly increased GSDMD in neutrophils in pustular psoriasis, we were constrained to studying the established PV animal model due to the current absence of a mature GPP animal model. This represents a limitation of our study.”

In summary, we appreciate the Reviewer’s comments and suggestions. We feel that the inclusion of new data addresses the concerns in a comprehensive manner and adds further support to our original conclusions. We hope you will now consider the revised manuscript worthy of publication in *eLife*.

References:

(1) Lian, N., et al., Gasdermin D-mediated keratinocyte pyroptosis as a key step in psoriasis pathogenesis. Cell Death & Disease, 2023. 14(9): p. 595.

(2) Han, J., et al., GSDMD (gasdermin D) mediates pathological cardiac hypertrophy and generates a feed-forward amplification cascade via mitochondria-STING (stimulator of interferon genes) axis. Hypertension, 2022. 79(11): p. 2505-2518.

(3) Lin, H., et al., Forsythoside A alleviates imiquimod-induced psoriasis-like dermatitis in mice by regulating Th17 cells and IL-17a expression. Journal of Personalized Medicine, 2022. 12(1): p. 62.

(4) Emami, Z., et al., Evaluation of Kynu, Defb2, Camp, and Penk Expression Levels as Psoriasis Marker in the Imiquimod‐Induced Psoriasis Model. Mediators of Inflammation, 2024. 2024(1): p. 5821996.

(5) Stackowicz, J., et al., Neutrophil-specific gain-of-function mutations in Nlrp3 promote development of cryopyrin-associated periodic syndrome. Journal of Experimental Medicine, 2021. 218(10): p. e20201466.

(6) Abram, C.L., et al., Distinct roles for neutrophils and dendritic cells in inflammation and autoimmunity in motheaten mice. Immunity, 2013. 38(3): p. 489-501.